# Kin discrimination allows plants to modify investment towards pollinator attraction

Rubén Torices [1,2], José M. Gómez[2] & John R. Pannell [1]

Pollinators tend to be preferentially attracted to large floral displays that may comprise more than one plant in a patch. Attracting pollinators thus not only benefits individuals investing in advertising, but also other plants in a patch through a 'magnet' effect. Accordingly, there could be an indirect fitness advantage to greater investment in costly floral displays by plants in kin-structured groups than when in groups of unrelated individuals. Here, we seek evidence for this strategy by manipulating relatedness in groups of the plant *Moricandia moricandioides*, an insect-pollinated herb that typically grows in patches. As predicted, individuals growing with kin, particularly at high density, produced larger floral displays than those growing with non-kin. Investment in attracting pollinators was thus moulded by the presence and relatedness of neighbours, exemplifying the importance of kin recognition in the evolution of plant reproductive strategies.

[1] Department of Ecology and Evolution, University of Lausanne, Lausanne CH-1015, Switzerland. [2] Departamento de Ecología Funcional y Evolución. Estación Experimental de Zonas Áridas, Consejo Superior de Investigaciones Científicas. Ctra. de Sacramento s/n, La Cañada de San Urbano, Almería E-04120, Spain. Correspondence and requests for materials should be addressed to R.T. (email: rubentorices@gmail.com)

The majority of plants rely on animal pollen vectors for their reproduction[1]. This interaction involves advertising a promised resource by the plant (e.g., nectar), and optimal foraging for the resource by the pollinator. Pollinators are more attracted to plants with many and large flowers[2], so that reproductive output increases with display size, at least when pollinators are scarce[3]. Because pollinator foraging behaviour is influenced not only by the floral signals made by individual plants[2, 3], but also by their spatial configuration[4–6], individual plants may benefit from their proximity to displaying neighbours through a so-called 'magnet-effect'[7–9].

The magnet-effect on an individual's inclusive fitness[10] ultimately depends on its relatedness to other individuals in its group. Plants growing in groups of relatives should cooperate in attracting pollinators more than individuals in groups of unrelated neighbours, because inclusive fitness is enhanced by benefits conferred upon kin[10]. This sort of pre-mating cooperation is well known in animals[11]. For example, in birds with lek mating behaviour, males display collectively to attract females more often when in groups of kin than non-kin[12, 13], and related *Drosophila* males fight less often with each other, and court females less aggressively, than do unrelated males[14]. In plants, we should expect individuals growing with kin to invest more in their floral display than those growing with unrelated neighbours.

Although populations can respond to kin selection in the absence of mechanisms allowing kin recognition[15], a capacity for kin discrimination should be advantageous in populations in which the relatedness among interacting individuals varies. Kin discrimination has indeed been observed in a phylogenetically broad range of organisms, including bacteria[16, 17], social amoebae[18], fungi[19] and animals[20, 21]. Increasing evidence also suggests that plants may recognise the identity of their neighbours and modify their phenotypes accordingly[22–25]. Plants of *Cakile edentula* increased root growth in the presence of non-kin compared with kin[26, 27]. *Arabidopsis thaliana* also responded to the identity of neighbours by increasing lateral root growth in the presence of root exudates from non-kin plants compared with kin[28]. Even seed emergence time, a trait with important effects on competitive ability[29, 30], has been found to vary as a function of the relatedness of neighbouring seeds: in *Plantago asiatica*, seeds accelerated their emergence in the presence of a competing species only if accompanied by kin[31]. Whereas all these examples involve responses to the context of resource competition, we ought to expect reproductive traits such as floral display similarly to depend on neighbour relatedness. However, to our knowledge, evidence for such sensitivity is yet to be reported.

Here, we asked whether individuals growing with kin invest more in their floral display than those growing with unrelated neighbours. We addressed this question by comparing the advertising effort of individuals of the plant *Moricandia moricandioides* growing with related vs. unrelated neighbours. *Moricandia moricandioides* is a self-incompatible annual-biennial plant with a patchy population structure and passive seed dispersal, and producing showy flowers pollinated by bees[32]. In the experimental neighbourhoods, individuals of *M. moricandioides* adjusted their floral displays to the composition of the social environment. As predicted, individuals growing with kin produced larger floral displays than those growing with non-kin. The effect was enhanced in dense clumps, indicating that not just the presence but also the number of relatives influences an individual's floral display. Our study demonstrates that, as shown for several plant competitive traits[7–9], and proposed by Hamilton[10] for reproductive traits, kin discrimination and kin selection may play a role in the evolution of plant-pollinator signalling.

## Results

**Advertising effort**. We raised individuals of *M. moricandioides* in pots in a glasshouse, either in groups of unrelated individuals sampled randomly from a single large source population ('non-kin' treatment), or in groups of individuals sharing the same mother but potentially different fathers ('kin' treatment). Groups comprised a focal individual surrounded by either three or six neighbours in the pot. As predicted, focal plants growing with kin invested significantly more in their floral display than those growing with unrelated neighbours, both in absolute and relative measures of allocation (Fig. 1a, Table 1). Both advertising components, number of flowers and mean petal mass, were consistently larger for focal plants growing with kin (Fig. 1b, c), a difference that was significant when allocation components were combined into a single index of advertising effort (Fig. 1a); the difference was particularly pronounced for plants growing in large groups (Fig. 1a). Our experiment thus revealed an important role of plasticity in floral traits apparently mediated by the social context.

**Social effects of resource competition**. To test whether the apparent effect of social context that we detected could be attributed to intraspecific competition instead, we compared focal plants growing in a competitive environment (i.e., within a group) with plants growing alone in pots of the same size, i.e., those having more soil resources. As expected, the biomass of solitary plants was significantly larger than that of plants growing in groups (Fig. 2a, Supplementary Table 1). Focal plants growing with six neighbours were also smaller than those growing with only three neighbours (Fig. 2a), although the difference in biomass was not affected by group relatedness (Fig. 2a; Table 1). Intraspecific competition is typically strong in plant populations[33]. However, we did not observe any difference in floral advertising between plants growing in competitive environments and those growing alone. By contrast, the floral display of focal plants growing with kin showed greater advertising effort than plants without competition (Fig. 1a). Resource depletion due to competition with neighbours was therefore not translated into reduced investment towards reproduction or attracting pollinators.

The observed greater advertising effort under the more competitive conditions of our experiment might also be attributable to allometric effects. To test this, we also included in our experiment single individuals growing in a soil volume equivalent to 1/4 or 1/7 of the group pots; these pots contained, on average, the same volume of soil as the equitable share of soil available to focal plants growing with three or six neighbours, respectively. Unsurprisingly, plants growing in smaller volumes of soil were significantly smaller than solitary plants growing in large pots (Fig. 2a), but they did not differ in their advertising effort (Fig. 1a). A decrease in resources therefore did not affect advertising effort per se. Importantly, rather than being larger, the advertising effort of solitary plants was significantly smaller than that of focal plants growing with kin (Fig. 1a; Supplementary Table 2). These contrasting allometric patterns indicate that the effects of social environment on reproductive allocation were different from those caused by resource deprivation.

## Discussion

The effect of the social environment on the floral display of *M. moricandioides* suggests that plants were capable of distinguishing between kin and non-kin neighbours. Plants have evolved sophisticated ways of recognising the identity of other organisms (reviewed in ref. [34]) in defending against pathogens[35], selecting hosts to parasitise[36], or in competing or sharing mycorrhizal

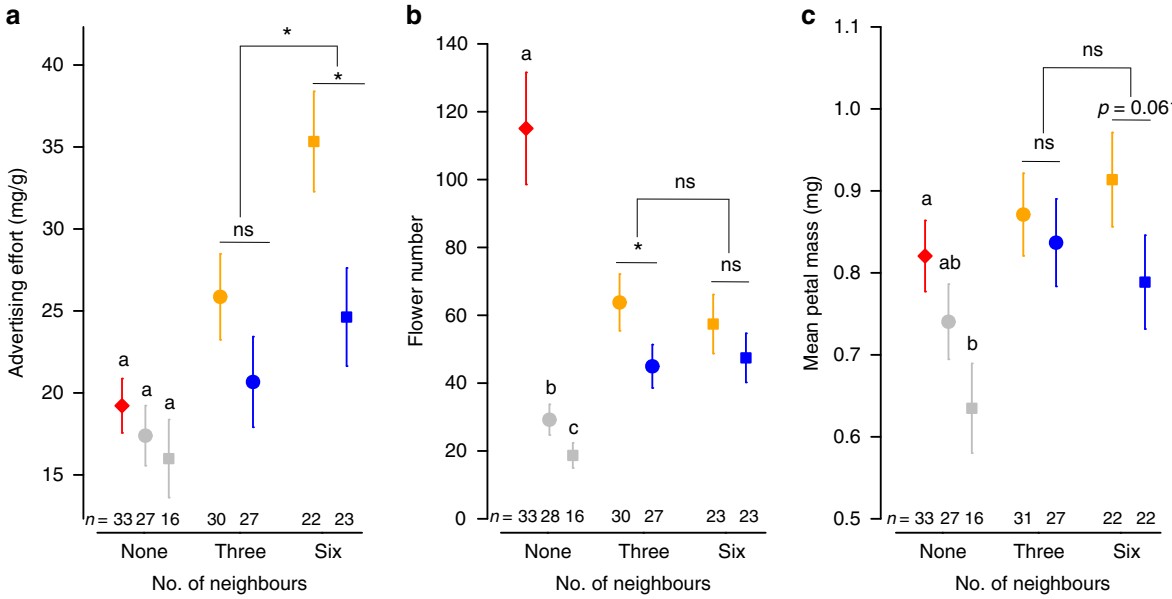

**Fig. 1** Effect of intraspecific social environment on floral display. Least-square means (± s.e.m.) of **a** advertising effort, **b** number of flowers, and **c** mean petal mass of focal plants. Focal plants were grown alone, with three neighbours, or with six neighbours, which were either kin (orange symbols) or non-kin (blue symbols). Solitary plants were grown in large (red diamonds), medium-sized (grey circles) and small pots (grey squares). Different letters indicate significant differences between solitary plants (GLMM test: P < 0.05, Supplementary Table 3), whereas differences between kin and non-kin treatments, and between levels of group size, are indicated above the symbols (GLMM test: ns, P > 0.05; *, P < 0.05, Table 1). P-values of comparisons between solitary plants vs. focal plants within groups are shown in Supplementary Table 2. All P-values were corrected for multiple comparisons using Holm's adjustment

| **Table 1 Effects of group size and neighbour relatedness on the performance of focal plants** | | | | |
|---|---|---|---|---|
| **Variables** | **n** | **d.f.** | **χ²** | **P** |
| Advertising effort | 102 | | | |
| Group size (G) | | 1 | 5.75 | **0.016** |
| Relatedness (R) | | 1 | 7.45 | **0.006** |
| G×R | | 1 | 0.96 | 0.326 |
| Number of flowers | 103 | | | |
| Group size (G) | | 1 | 0.05 | 0.827 |
| Relatedness (R) | | 1 | 5.38 | **0.020** |
| G×R | | 1 | 0.45 | 0.504 |
| Mean petal mass | 103 (293) | | | |
| Group size (G) | | 1 | 0.002 | 0.960 |
| Relatedness (R) | | 1 | 2.87 | **0.090** |
| G×R | | 1 | 1.09 | 0.297 |
| Individual biomass | 103 | | | |
| Group size (G) | | 1 | 8.09 | **0.004** |
| Relatedness (R) | | 1 | 0.95 | 0.330 |
| G×R | | 1 | 1.82 | 0.177 |
| Height | 103 | | | |
| Group size (G) | | 1 | 2.33 | 0.126 |
| Relatedness (R) | | 1 | 0.05 | 0.818 |
| G×R | | 1 | 0.49 | 0.486 |

The interaction between group size and neighbour relatedness, assessed using type-III tests, was not significant, so main effects were assessed using type-II tests of GLMMs. The relative position of the flower in the inflorescence was included as covariate for the models of petal mass. Plant family was included as random factor in all models, whereas individual plant was also included nested in plant family for the model fitting petal mass. Sample sizes (n) indicate the number of individuals and, for mean petal-mass, the number of flowers (in parenthesis). p-values below 0.1 are indicated in bold

networks and resources with neighbours[37]. The information transferred in these interactions may occur either below[34, 38] or above-ground[34, 39], but we remain largely ignorant of the mechanisms involved, as we do of those underlying the plasticity in advertising effort displayed by *M. moricandioides*. Despite this ignorance, our study extends empirical support for the capacity of plants to recognise kin, beyond its impact on vegetative competitive interactions[23–28, 38–40] to its role in shaping floral strategies of pollinator attraction.

The greater advertising effort of focal plants growing with kin than with non-kin is hard to explain without invoking kin selection. Kin selection has been invoked to account for resource allocation linked to plant competitive ability, including root growth[27, 38, 40] and the timing of seedling emergence[31]. The interpretation of some of these studies has been questioned, because a greater mean competitiveness of unrelated individuals may simply reflect either a greater variance in competitiveness (as a consequence of Jensen's inequality[41]), or the possibility that non-kin groups comprise one or more particularly competitive individuals[42, 43]. However, the focal plants in our experiment were neither smaller nor shorter when growing with non-kin (Fig. 2), so that explanations invoking competitive suppression are inadequate. On the contrary, in the more competitive six-neighbour environment, focal plants were smaller when growing with kin than with non-kin (Fig. 2a), whereas the effect of kin on advertising effort was greatest in this environment (Fig. 1a).

We might consider an increased investment in floral display as an altruistic rather than a mutually beneficial trait if it both benefits neighbouring plants and causes a reduction in the direct fitness of the focal plant[44]. We did not record the negative effects of advertising on individual plant fitness. However, colourful petals represent a potentially costly investment in biomass and costly pigments[45–47] that cannot be allocated to other reproductive functions[48, 49]. Accordingly, petal removal increased seed production, seed quality and seed germination in two species of *Nigella*[50, 51] and in three different daisy species[52–54], indicating that allocation to advertising structures might entail a cost on plant fitness. It is thus plausible that our results do represent a case of altruistic behaviour, but further work is required to demonstrate this.

Besides kin selection, plants surrounded by kin might also be selected to invest more in floral display to increase outcrossing. The efficiency of individual and group investment in floral display

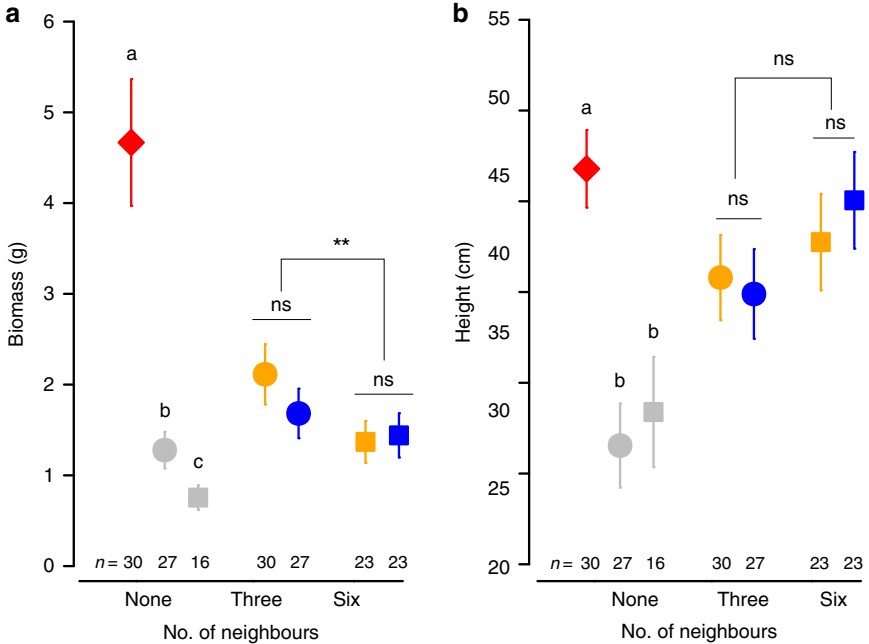

**Fig. 2** Effect of intraspecific social environment on plant size. Least-square means (± s.e.m.) of **a** plant above-ground biomass, and **b** plant height. Focal plants were grown alone, with three neighbours, or with six neighbours, which were either kin (orange symbols) or non-kin (blue symbols). Solitary plants were grown in large (red diamonds), medium-sized (grey circles) and small pots (grey squares). Different letters indicate significant differences between solitary plants (GLMM test: $P < 0.05$, Supplementary Table 3), whereas differences between kin and non-kin treatments, and between levels of group size, are indicated above the symbols (GLMM test: ns, $P > 0.05$; *, $P < 0.05$; **, Table 1). $P$-values of comparisons between solitary plants vs. focal plants within groups are shown in Supplementary Table 2. All $P$-values were corrected for multiple comparisons using Holm's adjustment

will thus be determined by how investment affects between-group vs. within-group competition for pollinators. If increased floral display leads to stronger competition for pollinators within a group, the cooperative effects of attracting pollinators to a patch could be offset by stronger competition among individuals for pollinators visiting their group[55]. For instance, plant height in *Silene tatarica*, a trait with a positive effect on pollinator attraction in the species, was positively selected at both the patch and individual levels[56], pointing to the potential importance of within-patch competition for pollinators. However, if important, we should expect to observe this effect in groups of both related and unrelated individual, so it would not explain the difference observed in our experiment between groups with contrasting relatedness. Nevertheless, the relative importance of kin selection and selection for outcrossing will ultimately also depend on how pollinators respond to variation in floral display both among groups and among individuals within groups. We also note that if the benefits of investing in advertising outweigh the costs at the individual level, too, the greater floral display of plants surrounded by relatives in *M. moricandioides* in our experiment might be viewed as mutualistic rather than as strictly altruistic.

Hypotheses to explain floral strategies have hitherto focussed on their potential direct effects on individual fitness[48, 49], but our results suggest that floral strategies have also been shaped by selection through their indirect effects on inclusive fitness, i.e., through the success of relatives. This possibility was foreshadowed by Hamilton[10] and has long been appreciated by zoologists[11–14]. Our study now provides evidence that investment towards attracting pollinators includes a component of neighbour relatedness, a necessary consequence of the fine spatial genetic structure of plant populations.

## Methods

**Experimental system and design.** We evaluated the effect of the intraspecific social environment on the floral display of *Moricandia moricandioides* (Boiss.)

Heywood (Brassicaceae), a self-incompatible annual weed of arid habitats of southeastern Spain that relies entirely on insect pollinators for its reproduction, mainly Antophorini bees[32, 57, 58]. Specifically, we manipulated the social context of focal plants, varying the number and relatedness of neighbours sharing the same pot. Focal plants had zero, three or six neighbours, and neighbours were either half-sib progeny of the same open-pollinated mother ('kin') or a random sample of progeny from other mothers in the population ('non-kin'). Each individual in non-kin pots came from a different family. All experimental neighbourhoods were established using seedlings as similar in size as possible to reduce size asymmetries that might affect competition hierarchies.

To obtain sufficient seedlings for our experiment, we sowed 10,000 seeds from 50 different mothers from the same population near Granada, Spain (37°07′11.8″N 3°43′47.2″W), collected in June 2015. From these 50 families, we selected 35 with the highest germination rates. Seedlings were re-potted into pots with 1.5 l of soil between the April 7th and 18th, 2016, when they had achieved a minimum size and had produced at least one leaf to reduce mortalities associated with seedling transplantation. All experimental neighbourhoods whose focal plants came from the same maternal family were re-potted the same day. We could establish all three- and six-neighbour groups in our design for 28 families; for the remaining seven families, there were insufficient seedlings for the six-neighbour treatment. Thus, we established 126 pots; half of them comprised kin neighbours, whereas the other half comprised non-kin neighbours. In total 70 and 56 of these pots, respectively, corresponded to three-neighbour and six-neighbour groups. Plants were grown at a constant 24 °C temperature at a 16:8 h day-night regime in a mixture of 50% sterilised compost and 50% topsoil (agricultural-garden soil), with a pH 7.4. All pots were irrigated with 135 ml of water daily.

To test whether allocation patterns observed in the more competitive conditions could be attributed to allometric changes due to reduction in plant size[59], we included in our experiment single individuals growing in smaller pots with less soil. Specifically, we tested whether a reduction in plant size results in a disproportional increase in allocation to pollinator attraction. Thus, in addition to solitary individuals growing in the large 1.5 l pots, we established them also in medium pots with 1/4 the soil volume (i.e., 0.375 l), as well as in small pots with 1/7 the volume (i.e., 0.214 l). Plants in smaller pots thus had access to soil of a volume $1/(n + 1)$ of that of plants growing with n neighbours. These solitary plants came from maternal families used as focal plants in the experimental neighbourhoods. Because we did not establish in the smallest pots those families for which was not possible to established the six-neighbour treatment, this part of our experiment involved a total of 98 solitary plants, 35 for large and medium pots and 28 for small ones. Overall, we established 224 pots and a total of 770 plants, with 224 focal and 546 neighbouring plants. Regardless of their size, pots were haphazardly distributed within the glasshouse and were displaced from other pots by at least 0.5 m to avoid contact or shading between plants in different pots.

The use of smaller pots did not lead to phenotypic differences that could be attributed to effects other than that of soil reduction, such us root space limitations[60–62]. Importantly, the size of individuals grown in smaller pots was proportional to the reduction on soil volume, indicating that pot size did not have any effect on plant size and number of flowers (Supplementary Fig. 1). Moreover, advertising effort was not significantly different between the three pot sizes (Fig. 1a), although mean petal mass was significantly smaller for plants in smaller pots (Fig. 1c).

Finally, we distinguished between the effects of resource limitation caused by competition and plastic responses not directly caused by competitive resource deprivation (and potentially attributable to social effects). Thus, we compared solitary plants in medium and small pots with focal plants in three- and six-neighbour groups. (Recall that these solitary plants were grown in the same soil volume as the equitable share of soil available to focal plants grown in groups had on average.) This comparison allowed us to infer a social response when focal plants growing within groups showed phenotypes different from those of solitary plants growing in effectively the same soil volume.

**Advertising effort**. We assessed plant attractiveness to pollinators in terms of relative allocation to floral display per plant ('advertising effort'), thus accounting for size differences between individuals and maternal families. Advertising effort was calculated as the total mass allocated to petal mass across all flowers after 26 days of flowering, divided by the plant biomass. We allowed focal plants to flower for 26 days, based on previous information on the mean flowering duration in natural populations of about 3–4 weeks (A. González-Megías, pers. comm.). After 26 days, all plants in each pot were harvested. We measured the total number of flowers produced and height of each individual. Plants were dried at 60 °C for at least 72 h and then weighed to the nearest 0.1 mg. We estimated mean mass allocation to petals for each individual, measuring petal mass for three different flowers per focal plant. We sampled three buds before anthesis, which were then fixed in FAA. In the laboratory, we extracted all the four petals by dissecting the flowers under the stereomicroscope. Petals were dried at 60 °C for at least 48 h and then weighed to the nearest 0.01 mg.

We estimated mean allocation to petal mass by accounting for the effect of flower position within the inflorescence; inflorescences of *M. moricandioides* are racemes of usually more than 20 flowers[32], which develop and open sequentially, with implications for flower size and within-flower allocation[63]. Accordingly, we recorded the relative position of each sampled flower within its inflorescence, and estimated marginal means using a linear mixed model, with the relative position in the inflorescence included in the model. Petal mass significantly declined from the bottom to the top of the inflorescences (Type-II main effects of the linear mixed model: $n = 1151$, $F_{1, 879} = 4.43$, $P = 0.035$; Supplementary Fig. 2).

**Statistical analyses**. We fitted generalised linear mixed models (GLMMs) to assess the effect of experimental neighbourhoods on floral display. First, we assessed the effect of relatedness and group size, including its interaction, in a joint analysis that included all focal plants growing in groups. Here, we fitted models for advertising effort, number of flowers, petal mass, and plant height, with maternal family of the focal plant included as a random blocking factor. For petal mass, we included in our model flower position in the inflorescence as an additional explanatory variable, as well as the identity of the individual plant as an additional random factor. All response variables were modelled assuming a Normal distribution, except for the number of flowers for which we assumed a negative binomial distribution. Biomass was log-transformed before fitting the model. Interactions were assessed using type-III tests; where these were not significant, main effects were subsequently assessed by type-II tests. We explored our results following recommendations of Zuur et al.[64] to assure that data met the assumptions of linear modelling. All GLMMs were fitted using the 'lme4' package[65] in R. Statistical differences between treatment levels were assessed by least-square mean differences, using the package 'lsmeans'[66]. P-values for post-hoc comparisons were corrected using Holm's adjustment.

To disentangle the effects of resource limitation caused by competition from plastic responses not directly caused by competitive resource deprivation, we compared focal plants growing in groups of three and six neighbours with their respective solitary control plants, i.e., medium pots for three-neighbour groups and small pots for six-neighbour groups. Here, we used the same approach described above, fitting GLMMs according to the same model specifications. Likewise, we fitted a GLMM for each response variable (advertising effort, number of flowers, petal mass, plant biomass, and plant height), though in this case the explanatory variable was the social environment with four treatment levels (neighbourhood of kin, neighbourhood of non-kin, solitary in big pot, solitary in a smaller pot a soil volume equivalent a plant's equitable share in the larger pots). Finally, we tested allometric responses on the allocation to floral display components by comparing only solitary plants growing in large, medium and small pots. As before, we fitted on GLMM for each response variable, but with pot size as the explanatory variable.

All our analyses were based on measurements of focal plants and their neighbours in which all individuals in the pot had survived for the whole experiment. We also excluded from our analysis all three- and six-neighbour groups in which at least one and two neighbouring plants failed to flower, respectively.

**Data availability**. The floral display and plant size data that support the findings of this study are available in figshare with the identifier (doi:10.6084/m9.figshare.5777430.v1)[67].

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

## Acknowledgements

We thank Sarah Beuvier, Yves Cuenot, Lucía DeSoto, Adela González-Megías, Félix Gréverath, Ana Machado, Nicolas Ruch for assistance with seed collection, plant growing and data collection. This work was part of a project that has received funding from the European Union's Horizon 2020 research and innovation programme under the Marie Skłodowska-Curie Grant Agreement No. 655653 and from Fundación BBVA (PR17-ECO-0021) and the Spanish Ministerio de Economia y Competitividad (CGL2017-86626-C2-1-P). Additional funding was provided by grant 31003A_163384 to JRP from the Swiss National Science Fundation.

## Author contributions

R.T., J.M.G. and J.R.P. conceived the project. R.T. performed the experiment and analysed the data. R.T., J.M.G. and J.R.P. wrote the manuscript.

## Additional information

**Competing interests:** The authors declare no competing interests.

