## [Peer Review File · Nature Communications]

Reviewers' comments:

Reviewer #1 (Remarks to the Author):

This is a very interesting study that reports a convincing role for kinship in modulating a plant's investment in floral display. The idea is a larger investment in floral display attracts pollinators not only to the focal plant but also to its neighbours, such that the investment has a greater inclusive-fitness payoff when the plant is surrounded by genetic relatives. This is tested in a very simple experimental set up in which potted *Moricandia moricandioides* are reared in the glasshouse alongside either maternal siblings or else nonrelatives (i.e. random members of the population), with the plants of the kin treatment exhibiting a significantly larger floral display, and this effect being more pronounced when the plants are reared in larger kin groups. I'm very supportive of publication.

Some comments:

1. Although the results of this paper are framed in terms of there being a role for kin selection, it is more specifically kin discrimination that appears to have been demonstrated here. And whilst it is useful to document evidence of kin selection in shaping a plant's reproductive biology I think it is vastly more exciting to have documented evidence for kin discrimination – I think many readers will be very surprised by the idea that a plant would be able to assess the extent to which it is surrounded by kin and adjust its growth accordingly. Although kin discrimination is finally touched on in line 127, I feel that it should be mentioned upfront (I would have incorporated it into the title).
2. Related to this, I feel it would be more useful to go beyond the couple of animal examples discussed in lines 59-62, and give a taxonomically broader overview of evidence for such kin discrimination effects: what is known about this in relation to fungi, microbes, and indeed other plants?
3. Line 31 talks about increased floral display benefiting the focal individual, but then line 33 describes this same phenotype as "potentially altruistic". This requires a little more thought, as altruism is formally defined in terms of a reduction in the focal actor's direct fitness (and adaptations that increase the fitness of both actor and a social partner are properly described as "mutually beneficial").
4. There seems to be a bit of a non sequitur in lines 38-40: "The effect was enhanced in large clumps, indicating that not just the presence but also the number of relatives influences an individual's floral display". Of course, the number of relatives in a clump can be increased whilst holding the total size of the clump constant (by reducing the number of non-relatives in the clump). So I think a slight rewording may be needed for precision.

As is my policy, I waive anonymity

Andy Gardner

Reviewer #2 (Remarks to the Author):

This is an unusually novel and exciting result. It adds an important target of kin selection - floral displays. The study was scientifically sound and presented effectively.

The authors reported that plants growing with kin produced larger floral displays than those growing with strangers. They considered alternative hypotheses that competition or differential access to resources rather than a 'magnet effect' could have produced these results and failed to find evidence that differential access to resources was responsible.

My only suggestion is that it would have been more convincing if below ground biomass or total biomass had been measured rather than only above ground biomass. Both height growth and above ground biomass may reflect competition for light while below ground biomass is arguably a better measure of resources that are available to the plant to invest in its floral display. So it seems possible that plants in more competitive environments could both grow taller with more above ground biomass and also have fewer resources that were available for reproduction.

Reviewer #3 (Remarks to the Author):

This manuscript clearly demonstrates that plants of *Moricandia moricandioides* increase their floral display when growing with relatives. This responses indicates kin recognition, and arguably increases the group fitness. The question is excellent, the experimental design is good and the figures are clear. While I have some suggestions, I think that this is an important result that is well worth publishing in a high profile journal.

1. Line 33-34, line 57. Altruism is defined as costly helping. suggests that investment in large floral displays is a potentially altruistic trait. However, I would argue that this is an example of helping that is favoured by direct benefits. Floral display benefits the individual and the group, while altruism benefits the group but disfavors the individual. See Aspí J, Jaakkola A, Tuomi J, Siikamaa P. 2003. Multilevel phenotypic selection on morphological characters in a metapopulation of *Silene tatarica*. *Evolution* 57:509–517 as well as Dudley 2015 Plant cooperation and File et al (2012. Fitness consequences of plants growing with siblings: reconciling kin selection, niche partitioning and competitive ability. *Proceedings of the Royal Society B: Biological Sciences* 279:209–218.) for reviews of multilevel selection studies. Lehmann and Keller models would indicate that while altruism is only favoured by kin selection, kin selection can favour other kinds of helping, increasing selection on that helping. Lehmann L, Keller L. 2006. The evolution of cooperation and altruism— a general framework and a classification of models. *Journal of Evolutionary Biology* 19:1365–1376.

2. 218-220-Advertising effect is a ratio variable – can the authors show that it is unrelated to plant size? Or use an additive approach (residuals, analysis of covariance) to

demonstrate advertising effect?

3. Lines 91-94 "However, we did not observe any difference in floral advertising between plants growing in competitive environments and those growing alone. By contrast, the floral display of focal plants growing with kin showed greater advertising effort than plants without competition (Fig. 1a; Table 1). " Formally, the test for kin effects should be kin vs non-kin at the same density, not a kin vs. control is statistically significant but non-kin vs. control is not statistically significant. The key table is Extended Table 2, which compares kin vs. non-kin traits at the same pot size and density. Table 1 is not helpful and not necessary. See point 4 for arguments about why the pot size controls are likely not controls.

4. Line 204 to 212 . The problem with pot size controls is that there are potential density responses, if pots are spaced differently in for the different sizes. Was aboveground density held constant while belowground resources were manipulated? Plant responses to density and associated red:far-red cues, including flowering, are large and well studied. How did the authors deal with confounding density cues for the controls.

5. Another potential benefit of greater floral display is attracting pollinators to increase mating with non-kin. Kin may be less competitive neighbours, but not mating with kin is likely favoured. Consequently, there could be kin recognition without kin selection, only individual selection. Without selection studies, we can't make the strongest argument for kin selection.

Point-by-point response to the referees' comments

We are grateful to the three reviewers for their constructive comments and suggestions for improving our manuscript. Below, we provide a point-by-point response to the referees' comments; their comments are cited in blue.

Reviewer #1 (Andy Gardner)

1. Although the results of this paper are framed in terms of there being a role for kin selection, it is more specifically kin discrimination that appears to have been demonstrated here. And whilst it is useful to document evidence of kin selection in shaping a plant's reproductive biology I think it is vastly more exciting to have documented evidence for kin discrimination – I think many readers will be very surprised by the idea that a plant would be able to assess the extent to which it is surrounded by kin and adjust its growth accordingly. Although kin discrimination is finally touched on in line 127, I feel that it should be mentioned upfront (I would have incorporated it into the title).

1. We have modified the title, introduced a new paragraph in the introduction and moved one paragraph in the discussion to emphasise kin discrimination more.

2. Related to this, I feel it would be more useful to go beyond the couple of animal examples discussed in lines 59-62, and give a taxonomically broader overview of evidence for such kin discrimination effects: what is known about this in relation to fungi, microbes, and indeed other plants?

2. We have introduced additional examples of kin discrimination in the new paragraph we have added in the new introduction (Page 3). Nevertheless, we have focused on examples of animals in this paragraph because they refer to kin discrimination in the context of pre-copulatory interactions. To our knowledge, this kind of pre-copulatory kin interactions is only known in animals.

3. Line 31 talks about increased floral display benefiting the focal individual, but then line 33 describes this same phenotype as “potentially altruistic”. This requires a little more thought, as altruism is formally defined in terms of a reduction in the focal actor's direct fitness (and adaptations that increase the fitness of both actor and a social partner are properly described as “mutually beneficial”).

3. We agree with this comment, and we have included a new paragraph in the discussion about it. However, we believe that it is appropriate to retain the use of “potentially altruistic” in the abstract, notably because investments in floral traits is usually made at the expense of other functions such as seed provisioning, i.e., there are likely to be costs to the behaviour we have recorded. More specifically,

resources allocated to advertising structures cannot be allocated to other limiting functions such as seed production. Colourful petals represent an investment of biomass and costly molecules as pigments, which should be maintained during flowering (Ashman & Baker 1992; Ashman & Schoen 1997; Méndez 2001). Therefore, it has generally assumed that resources allocated to advertising structures cannot be allocated to other reproductive functions imposing thus a fertility cost and consequently reducing fitness (Charlesworth & Charlesworth 1981; Sakai 1993). In a series of manipulative experiments with different species, Steffan Andersson has showed that petal removal increases seed production (Andersson 1993; 1999; 2000; 2005; 2008), indicating that petal production may reduce resources to seed development. Similar results were observed after nectar removal in *Blandfordia nobilis* (Pyke 1991).

In the system we describe, any disproportional increase on attractive structures could thus imply a reduction on the available resources for fruit and seed production and then in direct fitness. We have indeed found this to be so in *Moricandia moricandioides*: petal removal results in an increase in seed set compared to control unmanipulated flowers (unpublished data). Nevertheless, rather than cite these unpublished results, we have chosen to qualify our use of the term altruistic with “potentially”, because our study has not demonstrated the fitness costs explicitly. This is also examined briefly in discussing our results (Page 6).

4. There seems to be a bit of a non sequitur in lines 38-40: “The effect was enhanced in large clumps, indicating that not just the presence but also the number of relatives influences an individual’s floral display”. Of course, the number of relatives in a clump can be increased whilst holding the total size of the clump constant (by reducing the number of non-relatives in the clump). So I think a slight rewording may be needed for precision.

4. We have removed this sentence in the new abstract

Reviewer #2 (Remarks to the Author):

My only suggestion is that it would have been more convincing if below ground biomass or total biomass had been measured rather than only above ground biomass. Both height growth and above ground biomass may reflect competition for light while below ground biomass is arguably a better measure of resources that are available to the plant to invest in its floral display. So it seems possible that plants in more competitive environments could both grow taller with more above ground biomass and also have fewer resources that were available for reproduction.

5. We agree with the reviewer that having below-ground and total biomass would have been more convincing and might have provided insight into mechanisms by

which focal plants could have got extra resources. Unfortunately, measuring below ground biomass in this kind of experiment was not compatible with current experimental design. We needed roots were in contact with each other, so they ended up being totally entangled in a way that eventually precluded a correct individualization of roots within each pot.

As the reviewer indicates, in more competitive environments it is expected that plants grow taller and therefore with more above ground biomass, reducing thus the resources available for reproduction. However, we observed that advertising effort was in fact higher in the more competitive environments.

Reviewer #3 (Remarks to the Author):

1. Line 33-34, line 57. Altruism is defined as costly helping, suggests that investment in large floral displays is a potentially altruistic trait. However, I would argue that this is an example of helping that is favoured by direct benefits. Floral display benefits the individual and the group, while altruism benefits the group but disfavours the individual. See Aspí J, Jaakkola A, Tuomi J, Siikamaa P. 2003. Multilevel phenotypic selection on morphological characters in a metapopulation of *Silene tatarica*. *Evolution* 57:509–517 as well as Dudley 2015 Plant cooperation and File et al (2012. Fitness consequences of plants growing with siblings: reconciling kin selection, niche partitioning and competitive ability. *Proceedings of the Royal Society B: Biological Sciences* 279:209–218.) for reviews of multilevel selection studies. Lehmann and Keller models would indicate that while altruism is only favoured by kin selection, kin selection can favour other kinds of helping, increasing selection on that helping. Lehmann L, Keller L. 2006. The evolution of cooperation and altruism— a general framework and a classification of models. *Journal of Evolutionary Biology* 19:1365–1376.

6. Please, see our comment #3.

2. 218-220-Advertising effect is a ratio variable – can the authors show that it is unrelated to plant size? Or use an additive approach (residuals, analysis of covariance) to demonstrate advertising effect?

7. We did not observe any significant allometric pattern on advertising effort, nor any effect of including plant size as a covariate.

We defined advertising effort as the proportion of plant biomass allocated to petals. This variable allows for comparing plants with different sizes, as we expected larger plants to invest more resources into petals than smaller ones. In fact, our data address this question directly. Using only solitary plants, where there was no effect from different neighbourhoods, we observed that plants in larger pots and consequently with larger sizes (Fig. 2a) produced significantly more

flowers (Fig. 1b), and their petals were significantly heavier (Fig. 1c) than smaller plants that grew with fewer soil resources. However, there was no statistical difference in the advertising effort between them; larger plants showed slightly larger advertising effort than smaller plants (Fig 1a). Thus, our experimental data does provides no evidence that advertising effort might show strong allometric trends with plant size variation.

Additionally, we explored the effect of plant size on the advertising effort, including total plant mass as a covariate in all the GLMM(s) we fitted. We compared models with and without plant mass as a covariate (Table R1 in this document), and in any case the inclusion of plant mass as a covariate in the models produced an improvement of the model (Table R1).

Table R1. Comparison between GLMM fitting advertising effort as response variable and including (*with*) or not (*without*) plant mass as covariate. χ^2 and *P* were obtained using ‘anova’ function in R software

Model	Plant mass	AIC	χ^2	P
Table 1	Without	840.60		
	With	841.83	0.772	0.380
Supplementary table 1	Without	564.9		
	With	566.5	0.396	0.530
Supplementary table 3 (3 neighbours)	Without	917.09		
	With	918.76	0.323	0.570
Supplementary table 3 (6 neighbours)	Without	754.06		
	With	756.00	0.057	0.811

We also confirmed that including plant mass in the GLMMs did not lead to different patterns than our former models without plant mass (see below). We compared three types of models presented in the MS (Table 1, Supplementary Table 1 and 3). To facilitate the comparison, we included results from the manuscript and the results from the new GLMMs, including plant mass.

For Table 1:

Results without plant mass:

Variables	n	d.f.	χ^2	P
Advertising effort	102			
Group size (G)		1	5.75	0.016
Relatedness (R)		1	7.45	0.006
G x R		1	0.96	0.326

Results with plant mass:

Variables	n	d.f.	χ^2	P
Advertising effort	102			
Group size (G)		1	4.93	0.026
Relatedness (R)		1	7.34	0.006
G x R		1	0.79	0.374
Plant mass		1	0.77	0.379

For Supplementary table 1:

Results without plant mass:

	n	d.f.	χ^2	P
Advertising effort	76	2	1.35	0.508

Results with plant mass:

	n	d.f.	χ^2	P
Advertising effort	76	2	0.408	0.815
Plant mass		1	0.377	0.539

For Supplementary table 3:

Results without plant mass:

	3 neighbour groups			6 neighbour groups		
	n	χ^2	P	n	χ^2	P
Advertising effort	117	8.62	0.035	94	30.05	<0.001

Results with plant mass:

	3 neighbour groups			6 neighbour groups		
	n	χ^2	P	n	χ^2	P
Advertising effort	117	8.83	0.032	94	27.64	<0.001
Plant mass		0.31	0.578		0.06	0.812

In summary, there is no evidence that advertising effort was affected by plant mass in a way that might affect the observed patterns.

3. Lines 91-94 “However, we did not observe any difference in floral advertising between plants growing in competitive environments and those growing alone. By contrast, the floral display of focal plants growing with kin showed greater advertising effort than plants without competition (Fig. 1a; Table 1). “ Formally, the test for kin effects should be kin vs non-kin at the same density, not a kin vs. control is statistically significant but non-kin vs. control is not statistically significant. The key table is Extended Table 2, which compares kin vs. non-kin traits at the same pot size and density. Table 1 is not helpful and not necessary. See point 4 for arguments about why the pot size controls are likely not controls.

8. We agree with the reviewer that kin effects should be tested comparing kin vs. non-kin at the same density. For that reason, we showed that result first (first paragraph of results, Page 4); as the reviewer highlights, this result was provided in the former Extended Table 2. Following the reviewer’s suggestion, former Table 1 has been moved to Supplementary Material, and Extended Table 2 has been included in the main text as current Table 1.

However, we disagree with the second part of the reviewer’s comment. Solitary plants under different pot sizes, and thus with controlled soil resources, provided valuable controls. Using different amounts of soil resources, we obtained different plant sizes, which allowed us to test for potential allometric responses, but also to assess whether the responses observed could have been attributed to intraspecific competition (Page 4, last paragraph). See also our response #9.

4. Line 204 to 212. The problem with pot size controls is that there are potential density responses, if pots are spaced differently in for the different sizes. Was aboveground density held constant while belowground resources were manipulated? Plant responses to density and associated red:far-red cues, including flowering, are large and well studied. How did the authors deal with confounding density cues for the controls.

9. Aware that plants respond to density, in our experiment we separated all pots from each other by at least half a meter, so that plants from different pots could interfere with each other. Regardless of their size, pots were placed haphazardly within the glasshouse, and they were frequently redistributed onto different tables and positions within each table. We never place smaller pots together, so pot size was not a confounding factor and represent valid controls of the amount of soil resources available for the focal plant. We have expanded our account of these points in the revised manuscript (Page 9, end of first paragraph).

5. Another potential benefit of greater floral display is attracting pollinators to increase mating with non-kin. Kin may be less competitive neighbours, but not mating with kin is likely favoured. Consequently, there could be kin recognition without kin selection, only individual selection. Without selection studies, we can't make the strongest argument for kin selection.

10. Although we believe that kin selection is the most plausible explanation for our results, we have modified the title and introduced a new paragraph in the introduction to emphasise kin discrimination over kin selection, as explained above in response to comments of Reviewer 1.

References cited in this response

- Andersson, S. The Cost of Flowers in *Nigella degenii* Inferred from Flower and Perianth Removal Experiments. *Int. J. Plant Sci.* **161**, 903–908 (2000).
- Andersson, S. Floral Costs in *Nigella sativa* (Ranunculaceae): Compensatory Responses to Perianth Removal. *Am. J. Bot.* **92**, 279–283 (2005).
- Andersson, S. & Widén, B. Pollinator-mediated selection on floral traits in a synthetic population of *Senecio integrifolius* (Asteraceae). *Oikos* **66**, 72–79 (1993).
- Andersson, S. The Cost of Floral Attractants in *Achillea ptarmica* (Asteraceae): Evidence from a Ray Removal Experiment. *Plant Biol.* **1**, 569–572 (1999).
- Andersson, S. Pollinator and nonpollinator selection on ray morphology in *Leucanthemum vulgare* (oxeye daisy, Asteraceae). *Am. J. Bot.* **95**, 1072–8 (2008).
- Ashman, T. & Baker, I. Variation in Floral Sex Allocation with Time of Season and Currency. *Ecology* **73**, 1237–1243 (1992).
- Ashman, T. L. & Schoen, D. J. The cost of floral longevity in *Clarkia tembloriensis*: An experimental investigation. *Evol. Ecol.* **11**, 289–300 (1997).
- Charlesworth, D. & Charlesworth, B. The effect of investment in attractive structures on allocation to male and female functions in plants. *Evolution* **41**, 948–968 (1987).
- Méndez, M. Sexual mass allocation in species with inflorescences as pollination units: a comparison between *Arum italicum* and *Arisaema* (Araceae). *Am. J. Bot.* **88**, 1781–1785 (2001).
- Sakai, S. Allocation to attractive structures in animal-pollinated flowers. *Evolution* **47**, 1711–1720 (1993).

REVIEWERS' COMMENTS:

Reviewer #3 (Remarks to the Author):

The revised paper is much improved. The statistical and experimental design concerns I had have been addressed. The paper reads well, and demonstrates kin discrimination for reproductive allocation in response to relatedness of neighbours and density.

I would still argue that increased reproductive allocation in the presence of kin is either a behaviour that benefits the individual and the group (direct benefit helping) or a response to increase the likelihood of outcrossing, rather than costly helping (altruism) for the following reasons:

1. With most animal examples of altruism towards relatives, the helping is asymmetric: helpers at the nest help their relatives while sacrificing their own reproduction; Belding's ground squirrels give alarm calls that warn relatives but may attract predator attention. But in plant stands as in this example, the helping is symmetric, with both focal and neighbouring plants increasing allocation. So if higher levels of reproductive advertising are costly to an individual, as argued in lines 171, then all individuals in the population are simultaneously paying that cost, and are receiving the benefit. The argument for kin selection on competitive behaviour is that individuals sacrifice the potential high payoff they would gain from winning, and risk the major loss if their neighbours do compete – altruism as equivalent to prisoner's dilemma or a tragedy of the commons Dudley, Plant Cooperation. *AoB Plants*. 2015; 7: plv113.. With reproductive allocation, I am not seeing the equivalent prisoner's dilemma.
2. The arguments in lines 48-54 suggest that increasing reproductive allocation provides a direct benefit to the individual and the group.
3. Aspi et al 2003 found positive selection at the individual and group levels for floral height, a pollinator attraction trait.
4. It is analogous to the greater investment in mycorrhizae, a mutualistic interaction, for kin groups compared to stranger groups. File et al 2012 *Plant Kin Recognition Enhances Abundance of Symbiotic Microbial Partner*. *PlosOne*.

See the following paper for another apparent effect of kinship on a reproductive trait: Root competition influences pollen competitive ability in *Viola tricolor*: effects of presence of a competitor beyond resource availability?

Lankinen, Åsa LU (2008) In *Journal of Ecology* 96(4). p.756-765

Point-by-point response to the referees' comments

We are grateful to the reviewer and Editor for their constructive comments and suggestions for improving our manuscript. Below, we provide a point-by-point response to the referee's comments (in blue).

Reviewer #3

I would still argue that increased reproductive allocation in the presence of kin is either a behaviour that benefits the individual and the group (direct benefit helping) or a response to increase the likelihood of outcrossing, rather than costly helping (altruism) for the following reasons:

1. With most animal examples of altruism towards relatives, the helping is asymmetric: helpers at the nest help their relatives while sacrificing their own reproduction; Belding's ground squirrels give alarm calls that warn relatives but may attract predator attention. But in plant stands as in this example, the helping is symmetric, with both focal and neighbouring plants increasing allocation. So if higher levels of reproductive advertising are costly to an individual, as argued in lines 171, then all individuals in the population are simultaneously paying that cost, and are receiving the benefit. The argument for kin selection on competitive behaviour is that individuals sacrifice the potential high payoff they would gain from winning, and risk the major loss if their neighbours do compete – altruism as equivalent to prisoner's dilemma or a tragedy of the commons Dudley, Plant Cooperation. *AoB Plants*. 2015; 7: plv113. With reproductive allocation, I am not seeing the equivalent prisoner's dilemma.
2. The arguments in lines 48-54 suggest that increasing reproductive allocation provides a direct benefit to the individual and the group.
3. Aspi et al 2003 found positive selection at the individual and group levels for floral height, a pollinator attraction trait.
4. It is analogous to the greater investment in mycorrhizae, a mutualistic interaction, for kin groups compared to stranger groups. File et al 2012 Plant Kin Recognition Enhances Abundance of Symbiotic Microbial Partner. *PlosOne*.

We acknowledge that selection for increased outcrossing could in part explain the increase in attractiveness within dense groups, though it is difficult to account for the difference observed between groups of contrasting relatedness. Ultimately, the relative importance of kin selection and selection for outcrossing will also depend on the way pollinators discriminate among groups, in the first instance, and then among individuals within groups. We have added a paragraph that draws attention to these possibilities.